# Pathway to $O(\sqrt{d})$ Complexity bound under Wasserstein metric of flow-based models

## Abstract

We provide attainable analytical tools to estimate the error of flow-based generative models under the Wasserstein metric and to establish the optimal sampling iteration complexity bound with respect to dimension as $O(\sqrt{d})$. We show this error can be explicitly controlled by two parts: the Lipschitzness of the push-forward maps of the backward flow which scales independently of the dimension; and a local discretization error scales $O(\sqrt{d})$ in terms of dimension. The former one is related to the existence of Lipschitz changes of variables induced by the (heat) flow. The latter one consists of the regularity of the score function in both spatial and temporal directions.

These assumptions are valid in the flow-based generative model associated with the Föllmer process and 1-rectified flow under the Gaussian tail assumption. As a consequence, we show that the sampling iteration complexity grows linearly with the square root of the trace of the covariance operator, which is related to the invariant distribution of the forward process.

## 1 Introduction

The landscape of deep learning has been fundamentally reshaped by the emergence of powerful generative models, including Generative Adversarial Networks (GANs) (Goodfellow et al., 2014; Arjovsky et al., 2017), Variational Auto-encoders (VAEs) (Kingma & Welling, 2014; Kingma et al., 2019), and Normalizing Flows (Papamakarios et al., 2021; Wang et al., 2023; Wan & Wei, 2022), which have achieved remarkable success in a wide range of applications across modalities like images, audio, and text. These models are capable of learning complex data distributions, allowing them to generate high-quality samples (Achiam et al., 2023; Song et al., 2021).

Diffusion models (DM) are the state-of-the-art generative models, which can be analyzed via the SDE framework (Song et al., 2021). With the same forward and backward marginal as DM, flow-based models (Chen et al., 2023b;c) are generative models with deterministic flow given initial distribution, offering a strong basis for statistical inference. This unique feature makes them highly effective in applications such as image and audio synthesis, as well as density estimation (Cheng et al., 2024).

Early works on DMs and flow-based models provide reverse KL guarantees (Chen et al., 2023a; Benton et al., 2024; Conforti et al., 2025a; Li et al., 2024). However, for structured data, where the target typically lies on a compact sub-manifold (Tenenbaum et al., 2000; Bengio et al., 2017), the KL divergence between the backward process and the target is ill-defined. Therefore, one may turn to the analysis of flow-based models under the Wasserstein metric, and in this paper, we will consider the $W_2$ distance in Euclidean space, which is well-defined among distributions with finite second-order moments. One of the central difficulties in the analysis under the $W_2$ distance is the accumulation of local error in the Lyapunov-type estimate. This is in sharp contrast with KL-based analysis (Altschuler & Chewi, 2024; Zhu, 2025; Kim & Milman, 2012) which admits the **Girsanov**'s theorem (for instance, one in Chen et al. (2023a)) showing the constant scaling of the local error.

In light of this, **the main contribution of this paper** is to provide analytical tools that study the accumulation error along the sampling flow under the Wasserstein metric and hence ensure the optimal iteration complexity bound $O(\sqrt{d})$. More precisely, we first analyze the potential asymptotic

scaling of the truncation error in terms of the temporal variable and the ambient dimension. Then we bound the accumulation of error by the Lipschitz properties of the push-forward maps of the backward flow. As a justification, we illustrate the attainability of the assumptions by showing the optimal complexity bound in Föllmer flow under the Gaussian tail assumption. Such an assumption applies to both regular and singular targets (when early stopping technique (Lyu et al., 2022) is applied), extendable to infinite-dimensional settings, with further implications for Bayesian inverse problems.

## 1.1 RELATED WORK

Lipschitz changes of variables In the field of PDE, the Lipschitzness of transport maps was initiated by Caffarelli (2000), who constructed such maps between log-concave probability measures. Building on this, Colombo et al. (2017) developed global Lipschitz maps for compactly supported perturbations of log-concave measures. An alternative approach beyond optimal transport involves diffusion processes. By leveraging the maximum principle for parabolic PDEs, one can show that log-concavity is preserved along the associated diffusion semigroup (Kim & Milman, 2012). Mikulincer & Shenfeld (2023) obtained a sharper Lipschitz constant for measures with bounded support and Gaussian mixtures, improving Caffarelli's result. Based on this, Dai et al. (2023) assume a finite third moment and semi-log-convexity to construct a well-posed unit-time Föllmer flow whose terminal map is Lipschitz and pushes a Gaussian to a target measure in the unit time interval $[0, 1]$. Neeman (2022) relaxed Colombo's compact support requirement to boundedness, and Fathi et al. (2024) extended it to Gaussian in $\mathbb{R}^d$ and uniform spherical measures. Furthermore, Brigati & Pedrotti (2024) obtained the sharpest Lipschitz bound in this setting without controlling the third-order derivative tensor of potential $\nabla^3 W$. For clarity, we summarize the assumptions on target distributions and their Lipschitz constants in Table 2, with details in Appendix A. Despite these results shed light on potential minimal assumption for the convergence guarantee of flow-based models, in later context, we will demonstrate that estimation of the time derivative of velocity field $\partial_t V$ is also crucial on the pathway of optimal complexity bounds.

Continuous flow-based generative models Building on score-based (Song et al., 2021) and denoising diffusion models (Gao & Zhu, 2025), Salimans & Ho (2022) introduces stable parameterizations and a distillation method to reduce sampling steps while maintaining sample quality. Flow matching (FM) (Lipman et al., 2023) extends continuous normalizing flows (CNFs) (Chen et al., 2018) by training a neural ODE-parameterized vector field $v_\theta(x, t)$ to match a target velocity $v(x, t)$ along fixed probability paths, unifying diffusion and non-diffusion models for efficient and stable generation. Rectified flow (Liu et al., 2023; Rout et al., 2024) learns neural ODEs that transport distributions along nearly straight paths through iterative rectification processes, yielding deterministic couplings with reduced transport cost and enabling efficient one-step simulation. In addition, stochastic interpolants (Albergo et al., 2023; Albergo & Vanden-Eijnden, 2023) unify flow-based and diffusion-based methods to bridge arbitrary densities,

$$X_t = I(t, X_0, X_1) + \gamma(t)z, \quad t \in [0, 1], \tag{1}$$

recovering the Schrödinger bridge when the interpolant is optimized (Léonard, 2013). Recently, Flow Map Matching (FMM) (Boffi et al., 2025) has accelerated sampling by learning the two-time flow map of generative dynamics, thereby alleviating the computational cost associated with continuous models. Geng et al. (2025) connect one-step generative modeling to multiscale physical simulations via average velocity, achieving leading performance on ImageNet 256×256 without pre-training or distillation.

Convergence bounds Recent studies control the KL, $W_2$, and TV distances between the generative and target distributions to guarantee convergence and measure training discretization errors. Albergo & Vanden-Eijnden (2023) bounded the $W_2^2$ distance by $e^{1+2K} H(\hat{v})$ under the smoothness and Lipschitz assumptions, where $H(\hat{v})$ measures discrete velocity error. Albergo et al. (2023) derived KL-based perturbation bounds for CNF estimators, while FMM (Boffi et al., 2025) improved $W_2$ bounds for pre-trained models via Lagrangian and Eulerian distillation losses controlling the teacher-student Wasserstein gap. The estimation error of the FM has been analyzed for typical data distributions (e.g., manifold-supported) by Benton et al. (2023), and a nonparametric $\mathcal{O}(n^{-1/(d+5)})$ convergence rate under early stopping (Lyu et al., 2022) was established by Gao et al. (2024), where $n$ denotes the sample size. Subsequently, Cheng et al. (2024) showed JKO flows reach $\mathcal{O}(\epsilon^2)$ KL error in $N \lesssim \log(1/\epsilon)$ steps, extending to non-density cases and yielding mixed KL-$W_2$ guarantees. We summarize recent complexity results for diffusion models and flow-based models (under

Wasserstein distance) in Table 1. Detailed theorems appear in Lemmas A.10-A.16. In this work, we achieve an optimal dependence of $\mathcal{O}\left(\sqrt{d}\right)$ on the data dimension $d$ without the assumption of log-concaveness of the target.

Table 1: Complexity bounds for DM/flow-based models in $d$ dimensions: previous results vs. ours.

| Target $P_0$ | Complexity | Result |
|---|---|---|
| $P_0$ log-concave* | $\mathcal{O}\left(\frac{\sqrt{d}}{\epsilon_0}(\log\frac{d}{\epsilon_0})^2\right)$ | Gao & Zhu (2025) Tab. 1 |
| G-tail Ass.* | $\mathcal{O}\left(\frac{\sqrt{d}}{\epsilon_0}\log\frac{d^2}{\epsilon_0^2}\right)$ | Wang & Wang (2024) Cor. 3.5 |
| one-side Lip+weakly log-concave* | $\mathcal{O}\left(\frac{d^2}{\epsilon_0^2}\right)$ | Gentiloni-Silveri & Ocello (2025) Thm.3.5 |
| weakly log-concave* | $\mathcal{O}\left(\frac{d}{\epsilon_0^2}\right)$ | Bruno & Sabanis (2025) Thm.3.12 |
| **G-tail** Ass. 3.7 | $\mathcal{O}\left(\frac{\sqrt{d}}{\epsilon_0}\right)$ | This work Thm.3.15 |

* denotes works on diffusion models; $n$ is the sample size.

## 1.2 CONTRIBUTIONS

- We point out that the $W_2$-distance between the generative and target distributions is controlled by the Lipschitzness of the push-forward maps introduced by sampling flow. By providing concrete bounds on the Lipschitz coefficient, we obtain an explicit estimate of the accumulation error.

- While prior works often rely on smoothness or strict log-concavity, we adopt a general condition in applications-the Gaussian tail Assumption 3.7 to provide the well-posedness and Lipschitz regularity of Föllmer flow, with explicit, dimension-free Lipschitz bounds (Corollary 3.11 and Corollary 3.14).

- By leveraging the Gaussian tail Assumption 3.7 to obtain accurate upper bounds on the time derivative of velocity field $|\partial_t V|$ (Theorem 3.8), our framework avoids the need for end-point constraints or early stopping (Lyu et al., 2022), enabling training and sampling throughout the entire interval $t \in [0, 1]$. This framework naturally extends the $\mathcal{O}\left(\sqrt{d}\right)$ complexity results of the SDE flow to the deterministic flow, achieving even better complexity than previous approaches (Wang & Wang, 2024).

## 2 FLOW-BASED MODEL

We begin by introducing a unified formulation of flow-based generative models. This general framework allows the convergence analysis in Section 3 to apply seamlessly to both the Föllmer flow and more general sampling dynamics. Consider a continuous flow governed by a velocity field $V$ via the ODE[1]

$$\frac{d\overleftarrow{X}_t}{dt} = V\left(t, \overleftarrow{X}_t\right), \quad \overleftarrow{X}_0 = x, \quad t \in [0, 1]. \tag{2}$$

With the $N$ steps discretization in time, $0 = t_0 < t_1 < \cdots < t_N = 1$, the ODE (2) in each sub-interval $[t_n, t_{n+1}]$, can be interpreted as a local transport map,

$$T_n(\overleftarrow{X}_{t_n}) = \overleftarrow{X}_{t_{n+1}}. \tag{3}$$

The overall flow-based model $\overleftarrow{X}_1(x)$ is then obtained by the composition of transport maps

$$\overleftarrow{X}_1(x) = (T_{N-1} \circ T_{N-2} \circ \cdots \circ T_0)(x).$$

An approximation of $\overleftarrow{X}_1(x)$ can be interpreted as approximation of $\{T_n\}_{n=0}^{N-1}$ by $\{\widetilde{T_n}\}_{n=0}^{N-1}$. To quantify the error of the approximation, we denote the marginal distribution of the actual state $\overleftarrow{X}_{t_{n+1}}$

---

[1]We used the left arrow $\overleftarrow{\cdot}$ to represent its connections to the backward process in the score based model.

by $\overleftarrow{P}_{t_{n+1}}$, and $\overleftarrow{Q}_{t_{n+1}}$ of the approximated state. Correspondingly we have,

$$\overleftarrow{P}_{t_{n+1}} = (T_n)_{\#}(\overleftarrow{P}_{t_n}), \quad \overleftarrow{Q}_{t_{n+1}} = (\widetilde{T}_n)_{\#}(\overleftarrow{Q}_{t_n}). \tag{4}$$

**Föllmer flow**    For any $\varepsilon \in (0, 1)$, we consider a diffusion process $(\overrightarrow{X}_t)_{t\in[0,1-\varepsilon]}$ that gradually transforms the target distribution $\nu$ into a Gaussian $\mathcal{N}(0, C)$ over time by the following Itô SDE

$$d\overrightarrow{X}_t = -\frac{1}{1-t}\overrightarrow{X}_t dt + \sqrt{\frac{2C}{1-t}}dW_t, \quad \overrightarrow{X}_0 \sim \nu, \quad t \in [0, 1-\varepsilon], \tag{5}$$

where $W_t$ is a standard Brownian motion, $C$ is a symmetric, positive-definite covariance matrix. The transition probability distribution from $\overrightarrow{X}_0$ to $\overrightarrow{X}_t$ is given by

$$\overrightarrow{X}_t | \overrightarrow{X}_0 = x_0 \sim \mathcal{N}\big((1-t)x_0, t(2-t)C\big). \tag{6}$$

The marginal distribution flow $(\bar{p}_t)_{t\in[0,1-\varepsilon]}$ of the forward diffusion process satisfies the Fokker-Planck-Kolmogorov (FPK) equation in an Eulerian framework

$$\partial_t \bar{p}_t = \nabla \cdot \left( \bar{p}_t \cdot \frac{1}{1-t}[x + C\nabla \log \bar{p}_t(x)] \right) \quad \text{on } [0, 1-\varepsilon] \times \mathbb{R}^d, \quad \bar{p}_0 = \nu. \tag{7}$$

Then Föllmer flow is formally defined as the backward process of such a forward diffusion (5), while preserving the same marginal distributions in (7).

**Definition 2.1** (Föllmer flow in formal sense). *A Föllmer flow $(\overleftarrow{X}_t)_{t\in[0,1]}$ solves the IVP*

$$\begin{cases} \frac{d\overleftarrow{X}_t}{dt} = V(t, \overleftarrow{X}_t), \quad \overleftarrow{X}_0 \sim \gamma_C, \quad t \in [0, 1], \\ V(t, x) := \frac{1}{t}\left[x + S(t, x)\right], \quad \forall t \in (0, 1]; \qquad V(0, x) := \sqrt{C}\mathbb{E}_\nu[X], \end{cases} \tag{8}$$

$S(t, x) := C\nabla \log p_t(x)$ *is the score function with probability density* $p_t = \bar{p}_{1-t}$ *in forward FKP equation* (7). *We call* $V(t, x)$ *a Föllmer velocity field.*

Following (4), we define $\overrightarrow{P}_{t_n}$ as the marginal distribution of $\overrightarrow{X}_{t_n}$ in the forward diffusion process. Given the initial distribution $\overrightarrow{P}_0 = P_{\text{data}}$, then for all $t \in [0, 1]$, $\overleftarrow{P}_{t_n} = \overrightarrow{P}_{1-t_n}$.

In practice, the velocity field $V(1-t, x) = \frac{1}{1-t}[x + C\nabla \log \bar{p}_t(x)]$ is not available since no closed form expression of $\bar{p}_t$ is known. To this end, one approximates $V$ by a neural network $\widetilde{V}$. The network is trained by minimizing an $\mathbb{L}_2$ estimation loss,

$$\mathbb{E}_{\bar{p}_t(x)} \left\| \widetilde{V}(1-t, x) - \frac{1}{1-t}\left[x + C\nabla \log \bar{p}_t(x)\right] \right\|^2. \tag{9}$$

For simplicity, we introduce the notation $X_t := (1-t)X_0 + \sqrt{t(2-t)C}\,\mathcal{N}$, which shares the same marginal distribution as $\overrightarrow{X}_t$ in (6). Then the velocity field $V(1-t, x)$ can be expressed as a conditional expectation (Yubin et al., 2025),

$$V(1-t, X) := \frac{1}{1-t}\left[X + S(1-t, X)\right] = \mathbb{E}_{X_0|X_t}\left[ \frac{1}{1-t}X_t - \frac{X_t - (1-t)X_0}{(1-t)t(2-t)} \,\middle|\, X_t = X \right].$$

With an appropriate weight of the $t$-variable, the loss in (9) becomes an approximation of this conditional expectation via mean-squared prediction error,

$$\mathbb{E}_{X_0,\, N\sim\mathcal{N}(0,I_d),\, t}\left[ \lambda(t) \left\| \widetilde{V}\left(1-t, X_t\right) - \frac{1}{1-t}X_t + \frac{\sqrt{C}\mathcal{N}}{(1-t)\sqrt{t(2-t)}} \right\|^2 \right].$$

After training, with $\widetilde{V}(1-t, x)$, one can generate samples of the target distribution via an Euler-type discretization of the continuous-time process, starting from the Gaussian initialization $\gamma_C$,

$$\frac{d\overleftarrow{Y}_t}{dt} = \widetilde{V}(t_n, \overleftarrow{Y}_{t_n}), \quad \overleftarrow{Y}_{t_0} \sim \gamma_C, \quad t \in [t_n, t_{n+1}], \quad n = 0, 1 \ldots, N-1. \tag{10}$$

Note that (10) defines the transport map $\widetilde{T}_n$ for the learned Föllmer flow, governed by the approximate velocity field $\widetilde{V}(t_n, \overleftarrow{Y}_{t_n})$ over the sub-interval $[t_n, t_{n+1}] \subset [0, 1]$. Distribution of generation $\overleftarrow{Q}_t$ is then defined by (4).

**Well-posedness of Föllmer flow**    Under appropriate assumptions on the target distribution $\nu$, one can show the Föllmer flow being the time-reversal of the forward diffusion process (5). For instance, under third moment (Assumption 3.6), semi-log-convexity (Assumption A.19) and the structural assumptions (Assumption A.20) on $\nu$, Dai et al. (2023) studied the Föllmer flow in the case $C = I_d$, where the score function is given by

$$S(t, x) := \nabla \log \int_{\mathbb{R}^d} (2\pi(1 - t^2))^{-\frac{d}{2}} \exp\left(-\frac{|x - ty|^2}{2(1 - t^2)}\right) \nu(dy).$$

It can be shown that the velocity field $V$ is Lipschitz continuous in $x$ with a well-defined initial condition $V(0, x)$. By the Cauchy-Lipschitz theory (Ambrosio & Crippa, 2014), one can define a Lagrangian flow $(X_t^*)_{t \in [0,1]}$ governed by the well-posed ODE system,

$$dX_t^* = -V(1 - t, X_t^*)dt, \quad X_0^* \sim \nu, \quad t \in [0, 1],$$

sharing the same marginal distribution with (5).

In this work, we study the Föllmer flow with correlated Gaussian initial based on the Gaussian tail Assumption 3.7, and retain the spatially anisotropic noise assumption ($C \neq I_d$) to allow our theory to generalize to infinite-dimensional settings requiring compactification (Lim et al., 2025); We refer the reader to Theorem 3.8 for the regularity of the velocity field and Lemma 3.10 for the proof of well-posedness.

**General Notations**    Let $\gamma_C$ denote the density of $\mathcal{N}(0, C)$. For an $n \times n$ matrix $A$, the operator norm $\|\cdot\|$ is defined as

$$\|A\| = \sup_{v \neq 0} \frac{|Av|}{|v|} = \text{largest eigenvalue of } \sqrt{A^T A}.$$

For symmetric positive-definite $A$, define the weighted $\ell_2$ norm

$$|x|_A^2 := (A^{-1/2}x, A^{-1/2}x),$$

which reduces to the standard $\ell_2$ norm $|\cdot|$ when $A = I$. For a vector (matrix)-valued function $f(x)$,

$$|f|_\infty = \sup_x |f(x)|, \quad (\|f\|_\infty = \sup_x \|f(x)\|).$$

## 3    MAIN RESULTS

In this section, we present the main results. Our analysis begins with a general flow-based framework (not necessarily restricted to the Föllmer flow), through which we develop Wasserstein-based analytical tools that yield an optimal iteration complexity bound of $\sqrt{d}$. We then validate the assumptions and present the complexity results for the Föllmer flow and 1-rectified flow under the Gaussian tail assumption.

### 3.1    LIPSCHITZ CHANGES OF VARIABLES IMPLIES WASSERSTEIN BOUND OF FLOW-BASED MODELS

For the sake of compactness, we impose the following assumption on the second-order moment.

**Assumption 3.1** (Second moment)**.** *The data distribution has a bounded second moment, $M_2 := \mathbb{E}_{p_0}|x|^2 < \infty$. We further denote,*

$$M_0 = \max\{\text{Tr}(C), M_2\},$$

*relates to the maximum second-order moment, where $C$ is a symmetric, positive-definite covariance matrix.*

We consider a general covariance matrix $C$ to cover both the identity case $C = I_d$ and the correlated case $C \neq I_d$. In the main text, we primarily focus on the former, yielding $\text{Tr}(C) = d$ and thus $M_0 = \mathcal{O}(d)$ with dimension $d$. At the same time, we retain $C \neq I_d$ in the derivation to extend our theory to infinite-dimensional settings (Lim et al., 2025), with the general case further discussed in Appendix D for Bayesian inverse problems.

Next, we make three assumptions, each holding with some dimension-free constants. We regard these assumptions as generally valid, and under them, our convergence result Theorem 3.5 can be established.

**Assumption 3.2** (Lipschitzness of $T$). $\forall n = 0, \ldots, N-1$, $Lip(T_n) < \infty$, and $\prod_{j=0}^{n} Lip(T_j) < \infty$.

We will justify the attainability of the Assumption 3.2 in Corollary 3.11 by invoking the lipschitz property of the velocity field established in Theorem 3.8. Similar to Assumption 3.2 which imposes Lipschitz continuity of $T$, we also assume the Lipschitz continuity of $\widetilde{T}$ as stated below.

**Assumption 3.3** (Lipschitzness of $\widetilde{T}$). $\forall n = 0, \ldots, N-1$, $Lip(\widetilde{T}_n) < \infty$, and $\prod_{j=0}^{n} Lip(\widetilde{T}_j) < \infty$.

We will verify Assumption 3.3 in Corollary 3.14 by leveraging the lipschitz property of the learned velocity field stipulated in Assumption 3.13. The final assumption concerns the local discretization error between $T$ and $\widetilde{T}$ at each time step $h$, as described below.

**Assumption 3.4** (Accuracy of approximation). *There exists constants $\overline{K}, \overline{K_1}, \overline{K_2}, \epsilon$, such that*

$$\sqrt{\mathbb{E}_{x \sim \overleftarrow{P}_{t_n}} |\widetilde{T}_n(x) - T_n(x)|^2} \leq h \left( \left( \overline{K}\sqrt{M_0} + \frac{\overline{K_1}}{\sqrt{1 - t_n^2}} + \overline{K_2} \right) h + \epsilon \right),$$

*with time step size $h = t_{n+1} - t_n$.*

This scaling follows since

$$\widetilde{T}_n(x) - T_n(x) = h\big(V(x) - \widetilde{V}(x)\big) + \mathcal{O}(h^2),$$

as verified in the Föllmer case Theorem 3.15. The term $\mathcal{O}(h)$ reflects the $\epsilon$-accuracy of the learned velocity $\widetilde{V}(x)$ (Assumption 3.12), while the term $\mathcal{O}(h^2)$ stems from the Taylor expansion of $T_n(x)$ over $[t_n, t_{n+1}]$ and depends on its regularity, possibly also on ambient dimension $d$ and time $t$.

Next, we outline the core proof strategy of this work. The key step is to demonstrate the Lipschitz continuity of both the original and discretized flows, which is critical for guaranteeing the convergence of flow-based generative models.

**Theorem 3.5.** *Assume that the target distribution satisfies Assumption 3.1 and follows Lipschitzness Assumption 3.2, 3.3, and approximation error Assumption 3.4. With constant step size $h$, the Wasserstein-2 distance between the target distribution $\overrightarrow{P}_0 = \overleftarrow{P}_1$ and the generation $\overleftarrow{Q}_1$ is bounded as,*

$$\mathcal{W}_2(\overleftarrow{P}_1, \overleftarrow{Q}_1) \leq \left( \prod_{j=0}^{N-1} Lip(\widetilde{T}_j) \right) \mathcal{W}_2(\overleftarrow{P}_0, \overleftarrow{Q}_0)$$

$$+ h \sum_{k=0}^{N-2} \prod_{j=k}^{N-2} Lip(\widetilde{T}_j) \left( \left( \overline{K}\sqrt{M_0} + \frac{\overline{K_1}}{\sqrt{1 - t_j^2}} + \overline{K_2} \right) h + \epsilon \right). \tag{11}$$

Proof see Appendix B.1.

This result shows that the first term in the bound scales the initial discrepancy $\mathcal{W}_2(\overleftarrow{P}_0, \overleftarrow{Q}_0)$ by the product of Lipschitz constants $\left( \prod_{j=0}^{N-1} \text{Lip}(\widetilde{T}_j) \right)$, and the second term $\left( \left( \overline{K}\sqrt{M_0} + \frac{\overline{K_1}}{\sqrt{1-t_j^2}} + \overline{K_2} \right) h + \epsilon \right)$, captures accumulated discretization errors (Assumption 3.4) and a local discretization error scales $\mathcal{O}(\sqrt{M_0})$, yielding the $\mathcal{O}(\sqrt{d})$ dependence in the isotropic case $C = I_d$. Similar results (17) and (18) are listed in Proposition A.10 and Proposition A.13, while the precise scaling of the second term remains unspecified. To be noted, in the limit of $h \to 0$, $h \sum_{k=0}^{N-2} \frac{1}{\sqrt{1-t_k^2}} \to \frac{\pi}{2}$.

Notably, Theorem 3.5 is of general validity: it applies to all flow-based models and their discrete counterparts satisfying the relevant assumptions, and is not limited to the Föllmer case.

## 3.2 ANALYSES OF FÖLLMER FLOW UNDER GAUSSIAN TAIL ASSUMPTION

In this section, we focus on the Föllmer flow and derive the main convergence result based on Theorem 3.5 through Lipschitz changes of variables, which plays a central role in our analysis.

**Assumption 3.6** (Third moment). *The data distribution has a bounded third moment, i.e. $\mathbb{E}_{p_0}|x|^3 < \infty$.*

We note that the third-moment assumption 3.6 is only required to ensure well-posedness of Föllmer flow at $t = 0$ in the proof of Lemma 3.10 (see Appendix B.3). For our complexity bound, the second moment Assumption 3.1 is sufficient.

Our analysis is based on the following key assumption that the tail distribution of the target is similar to a Gaussian distribution with covariance matrix $A$.

**Assumption 3.7** (G-tail). *The density of target distribution $\bar{p}_0 \in C^2(\mathbb{R}^d)$ and has the following tail decomposition:*

$$\bar{p}_0(x) = \exp\left(-\frac{|x|_A^2}{2}\right)\exp(h(x)),$$

*where there are independent of dimension constants such that,*

*(i) $A$ is a symmetric, positive-definite matrix which can be simultaneously diagonalized with $C$, and*

$$\|A\| < \infty, \quad \|C\| < \infty, \quad \|AC^{-1}\| < \infty, \quad \|CA^{-1}\| < \infty.$$

*(ii) the remainder term $h$ follows*

$$|\sqrt{C}\nabla h|_\infty < \infty, \quad \|C\nabla^2 h\|_\infty < \infty.$$

The Gaussian tail Assumption 3.7 generalizes the log-concavity condition in Ding et al. (2023); Gao et al. (2024) to heavier-than-sub-Gaussian tails while ensuring sufficient decay for well-posedness and convergence. Although stronger than the weak semi-log-concavity assumption of Chaintron et al. (2025); Bruno & Sabanis (2025), it yields sharper guarantees: weak semi-log-concavity implies $O(d)$ sampling complexity, whereas the Gaussian tail assumption achieves $O(\sqrt{d})$ scaling in a non-log-concave setting and also accommodates realistic distributions such as early stopping, see (16).

The following theorem bounds the Lipschitz constant and the time derivative of the Föllmer velocity field in (8) under the Gaussian tail Assumption 3.7, supporting the Lipschitz changes of variables in Corollary 3.11 and convergence rate in Theorem 3.15.

**Theorem 3.8** (Regularity of the velocity field). *The Gaussian tail Assumption 3.7 implies the Föllmer velocity field $V(t, \cdot)$ has the following regularity properties:*

$$\begin{aligned}
|V(t,x)| &\leq K_0 + K_2 t|x|, & \forall t \in [0,1], \\
\|\nabla V(t,\cdot)\|_\infty &\leq (K_1 + K_2)t, & \forall t \in [0,1], \\
|\partial_t V(t,x)| &\leq K_5|x| + \frac{K_6}{\sqrt{1-t^2}} + K_7, & \forall t \in [0,1),
\end{aligned} \tag{12}$$

*where the coefficients are dimension-free constants, given explicitly in Table 3 of Appendix B.2.*

To handle the blow-up of $|\partial_t V(t,x)|$ near $t = 0, 1$, Ding et al. (2023) restrict $t$ to $[\delta, 1-\delta]$. In particular, Gao et al. (2024) shows Lipschitz continuity of $V$ in $t$ over $[0, 1-\delta_0]$ with constant scaling as $\mathcal{O}(\delta_0^{-2})$. In contrast, under our Gaussian tail assumption, the control over the second derivative of the tail allows us to bound $|\partial_t V(t,x)|$ using techniques such as the Brascamp-Lieb inequality (Brascamp & Lieb, 1976). This analysis reveals that the term $\frac{1}{\sqrt{1-t^2}}$ is integrable on $[0,1]$, thus posing no obstacle to convergence, allowing training and sampling over the full interval $t \in [0,1]$. More importantly, this approach, to our knowledge, is the first to yield the improved $\mathcal{O}(\sqrt{d})$ complexity bound, as formally stated in Corollary 3.16

Detailed proof of Theorem 3.8 is provided in Appendix B.2.

**Remark 3.9.** *Motivated by the averaged-velocity construction in MeanFlow (Geng et al., 2025), we introduce an analogous notion for the Föllmer flow and define the* averaged Föllmer velocity *as*

$$\overline{V}(x,r,t) := \frac{1}{t-r}\int_r^t V(\tau,x)\,d\tau.$$

*Under the regularity condition* (12) *satisfied by the Föllmer velocity field, a direct calculation gives the uniform bound*

$$\left|\overline{V}(x,r,t)\right| \;\leq\; K_0 \;+\; \frac{t+r}{2}\,K_2\,|x|,$$

*demonstrating that the averaged Föllmer velocity preserves the same linear growth property as the original velocity field.*

Under the preceding assumptions and analysis, we establish the well-posedness of the Föllmer model $(\overleftarrow{X}_t)_{t\in[0,1]}$ in the following lemma.

**Lemma 3.10** (Well-posedness). *Suppose that the third moment Assumption 3.6 and the Gaussian tail Assumption 3.7 hold. Then the Föllmer velocity field is well-defined at the $t=0$, in the sense that*

$$V(0,x) := \lim_{t\to 0} V(t,x) = \lim_{t\to 0} \frac{x+S(t,x)}{t} = \sqrt{C}\,\mathbb{E}_{\bar{p}_0}[X]. \tag{13}$$

*Consequently, the Föllmer flow $(\overleftarrow{X}_t)_{t\in[0,1]}$ is a unique solution to IVP* (8). *Moreover, the push-forward measure satisfies*

$$\gamma_C \circ (\overleftarrow{X}_1)^{-1} = \bar{p}_0.$$

Proof see Appendix B.3. Under Assumption 3.7, we now establish the Lipschitz property of the continuous flow $(\overleftarrow{X}_t)_{t\in[0,1]}$.

**Corollary 3.11** (Lipschitzness of continuous flow). *If $\bar{p}_0$ follows the Gaussian tail Assumption 3.7, then the Föllmer flow $(\overleftarrow{X}_t)_{t\in[0,1]}$ is Lipschitz with a dimension-free constant, more precisely,*

$$Lip(\overleftarrow{X}_1(x)) \leq \|\nabla \overleftarrow{X}_1(x)\|_{op} \leq \exp\left(\frac{K_1+K_2}{2}\right). \tag{14}$$

Proof see Appendix B.4. Bound like (14) can also be achieved in Caffarelli (2000); Colombo et al. (2017); Kim & Milman (2012); Mikulincer & Shenfeld (2023); Brigati & Pedrotti (2024) under various assumptions, as detailed in Appendix A. In general, the constants involved are dimension-free.

To analyze the stability and convergence of the discrete flow, we first assume the following bound on the velocity field approximation error at the discretization points.

**Assumption 3.12** (Accuracy of the learned velocity field). *For each time discretization point $t_n$, the accuracy of learned velocity $\widetilde{V}(t_n,x)$ approximates the true velocity field $V(t_n,x)$ with uniformly bounded error in expectation:*

$$\mathbb{E}_{\overrightarrow{P}_{1-t_n}} |V(t_n,x) - \widetilde{V}(t_n,x)|^2 \leq \epsilon^2.$$

Next, we assume that the learned velocity field inherits the regularity of the continuous flow under the Gaussian tail Assumption 3.7.

**Assumption 3.13** (Regularity of the learned velocity field). *Assume the learned velocity field $\widetilde{V}(t,x)$ follows*

$$\|\nabla\widetilde{V}(t_n,\cdot)\|_\infty \leq (K_1+K_2+K_8)t_n$$

*for some positive constant $K_8$.*

Although the bound may not be small in general, the Assumption 3.13 is essential for our theoretical analysis and remains reasonable. The learned velocity field $\widetilde{V}(t_n,x)$ is trained to approximate the true velocity field $V(t_n,x)$ in Assumption 3.12, which satisfies the required regularity (see Theorem 3.8); Moreover, neural networks generally inherits the smoothness and controlled growth induced by the architecture and training process, which prevents uncontrolled behavior in practice. Assumption 3.13 can further be relaxed in the temporal $t$ direction to require only that the total discrete-time sum of the score gradient is bounded; see Remark B.1 in Appendix B.7.

We subsequently establish the Lipschitz property of the discrete flow $(\overleftarrow{Y}_t)_{t\in[0,1]}$ under Assumption 3.13.

**Corollary 3.14** (Lipschitzness of discrete flow). *The regularity of learned velocity field Assumption 3.13 implies the Lipschitz property of the learned flow $(\overleftarrow{Y}_t)_{t \in [0,1]}$ with a dimension-free constant, such that*

$$Lip(\overleftarrow{Y}_1(x)) \leq \|\nabla \overleftarrow{Y}_1(x)\|_{op} \leq \exp\left(\frac{K_1 + K_2 + K_8}{2}\right),$$

Proof see Appendix B.5.

## 3.3 Main Convergence Theories

With the Lipschitz properties of the flow established (see Corollary 3.11 and Corollary 3.14), we next quantify how these bounds propagate through the discrete dynamics. Building on Theorem 3.5, the following theorem provides a convergence result in Föllmer flow case.

**Theorem 3.15.** *Suppose that the third moment Assumption 3.6, the Gaussian tail Assumption 3.7, the accuracy and regularity assumptions 3.12- 3.13 on the learned velocity field hold. Using the Euler method for the Föllmer flow with uniform step size $h = t_{n+1} - t_n \leq 1$ ensures $\sqrt{M_0}$ convergence between the target data distribution and the generated distribution:*

$$\mathcal{W}_2(\overrightarrow{P}_0, \overleftarrow{Q}_1) \leq \exp\left(\frac{K_1 + K_2 + K_8}{2}\right)\left(\sqrt{3}\left(K_5\sqrt{M_0} + K_9\right)h + 2\epsilon\right). \quad (15)$$

*where $K_1, K_2, \ldots, K_9$ are dimensionless constants defined in Theorem 3.8 and Assumption 3.13, with explicit expressions given in Table 3. Furthermore, with the covariance of base distribution $C = I_d$ in the Assumption 3.1, $\mathcal{W}_2(\overrightarrow{P}_0, \overleftarrow{Q}_1) = \mathcal{O}(\sqrt{d}\,h + \epsilon)$.*

Proof see Appendix B.6. Note that the first term in Theorem 3.5, stemming from the time-propagating discrepancy of the semigroup maps, vanishes in Theorem 3.15 because the Föllmer flow $(\overleftarrow{X}_t)_{t \in [0,1]}$ is well-posed at $t = 0$, giving $\mathcal{W}_2(\overleftarrow{P}_0, \overleftarrow{Q}_0) = 0$. Thus, only the accumulated discretization error remains, corresponding to the second term in Theorem 3.5.

**Corollary 3.16.** *To reach a distribution $\overleftarrow{Q}_1$ such that $\mathcal{W}_2(\overrightarrow{P}_0, \overleftarrow{Q}_1) = \mathcal{O}(\varepsilon_0)$ with uniform step size $h = t_{n+1} - t_n \leq 1$ requires at most:*

$$h = \mathcal{O}\left(\frac{\epsilon_0}{\sqrt{M_0}}\right), \quad N = \frac{1}{h} = \mathcal{O}\left(\frac{\sqrt{M_0}}{\epsilon_0}\right),$$

*and Assumption 3.12 to hold with $\epsilon = \mathcal{O}(\epsilon_0)$. Furthermore, $N = \mathcal{O}\left(\frac{\sqrt{d}}{\epsilon_0}\right)$ under the Assumption 3.1 with $C = I_d$.*

The complexity bound established in Corollary 3.16 grows linearly with the square root of the trace of the forward process's covariance operator, independent of dimension, and thus extends naturally to infinite-dimensional generative models. An illustrative case is provided in Appendix D, where we consider Bayesian inverse problems in function spaces. Proposition 6 in Gao et al. (2025) establishes that for the standard Gaussian as target distribution, $\mathcal{O}(\sqrt{d})$ complexity bound is optimal. This indicates that our $\sqrt{d}$ dependence stems from intrinsic Gaussian concentration, making the dimensional scaling fundamental rather than algorithm-induced. Notably, in efforts to obtain complexity bounds under assumptions more general than log-concavity, recent works (Bruno & Sabanis, 2025) derived an $\mathcal{O}(d)$ bound using the weakly log-concave assumption (Conforti, 2024; Conforti et al., 2025b), while (Gentiloni-Silveri & Ocello, 2025) obtained an $\mathcal{O}(d^2)$ bound under the similar assumption. These related works are summarized in Table 1.

Since the probabilistic ODE (Prob ODE) (Song et al., 2021; Gao & Zhu, 2025) can be viewed as a time-rescaled Föllmer flow, the result of Corollary 3.16 also implies that our method improves the computational complexity of the Prob ODE compared to Wang & Wang (2024). We will provide a detailed discussion in Appendix C.

We further verified that our method extends to the 1-rectified flow setting (Liu et al., 2023; Rout et al., 2024). In particular, it applies to the interpolation paths used in the flow built by the first step rectification over independent coupling prior to the recursive construction, and retains the same $\mathcal{O}(\sqrt{d})$ complexity stated in Corollary 3.16. The proof is deferred to Appendix B.8.

### 3.4 CONVERGENCE UNDER BOUNDED-SUPPORT ASSUMPTION

Real-world data often lie on low-dimensional manifolds, where the distribution is not absolutely continuous with respect to Lebesgue measure in the ambient dimension, and therefore KL bounds may diverge (Pidstrigach, 2022). This motivates the adoption and study of the manifold assumption (De Bortoli, 2022; Yubin et al., 2025), which, under compactness, entails the following bounded-support assumption.

**Assumption 3.17** (Bounded-support assumption). *Suppose distribution $p_0$ has compact support with $\mathrm{Diam}(\mathrm{Supp}(p_0)) \leq R$ for some constant $R > 0$.*

Let $q_\sigma = \exp\left(-\frac{|x|^2}{2\sigma^2}\right) * q_0$, where $q_0$ satisfies the bounded-support Assumption 3.17. Consider $g(x) = \log q_\sigma(x) + \frac{|x|^2}{2\sigma^2}$, inspired by similar results in (De Bortoli, 2022; Mooney et al., 2025; Wang & Wang, 2024), we have

$$|\nabla g|_\infty \leq \frac{R}{\sigma^2}, \quad \|\nabla^2 g\|_\infty \leq \frac{2R^2}{\sigma^4}. \tag{16}$$

Set $0 = t_0 < t_1 < \cdots < t_N = 1 - \delta$ as the discretization points, where the early stopping (Lyu et al., 2022) coefficient $\delta \ll 1$. By expressing the distribution of the forward process of Föllmer flow at stopping time $\delta$ in the form $q_\sigma$, we obtain the correspondences

$$\sigma^2 \longleftrightarrow 1 - (1-\delta)^2, \quad q_0(x) \longleftrightarrow \frac{1}{1-\delta}\overrightarrow{P}_0(\frac{1}{1-\delta}x).$$

Then by Theorem 3.8, we get the following Lipschitz bound of the velocity field under Assumption 3.17.

**Corollary 3.18.** *Suppose that the bounded-support Assumption 3.17 holds. Taking $C = I_d$ in (8), and $A = (1 - (1-\delta)^2)I_d$ in Assumption 3.7, then for all $t \in [0, 1-\delta]$,*

$$|V(t,x)| \leq K_0^* + K_2^* t|x|,$$
$$\|\nabla V(t,\cdot)\|_\infty \leq (K_1^* + K_2^*)t,$$
$$|\partial_t V(t,x)| \leq K_5^*|x| + \frac{K_6^*}{\sqrt{1-t^2}} + K_7^*,$$

*where coefficients are defined in Table 4 of Appendix B.2.*

The proof parallels the corollary in Wang & Wang (2024). Using the Lipschitz bound from Corollary 3.18, we obtain a bounded-support-version $W_2$ bound by tracking the constants in Theorem 3.15.

**Theorem 3.19.** *Suppose that the bounded-support Assumption 3.17 and the accuracy and regularity Assumptions 3.12, 3.13 hold. Take $C = I_d$, $\delta \ll 1$, then we have*

$$\mathcal{W}_2(\overrightarrow{P}_\delta, \overleftarrow{Q}_{1-\delta}) \leq \exp\left(\frac{3R^2}{2\delta^2} + \frac{1}{2\delta} + \frac{K_8}{2}\right)\left(\sqrt{3}\left(K_5^*\sqrt{M_0} + K_9^*\right)h + 2\epsilon\right),$$

*where $K_5^*$ and $K_9^*$ are dimension-free constants, whose explicit forms given in Table 4, and the constant $K_8$ is defined in Assumption 3.13.*

With the result in Theorem 3.19, we can directly compute the complexity bound under the bounded-support assumption with early stopping technique.

**Corollary 3.20.** *With $R$ and $\delta$ fixed, achieving a distribution $\overleftarrow{Q}_{1-\delta}$ such that $\mathcal{W}_2(\overrightarrow{P}_\delta, \overleftarrow{Q}_{1-\delta}) = \mathcal{O}(\epsilon_0)$ requires at most: $N = \mathcal{O}\left(\frac{\sqrt{d}}{\epsilon_0}\right)$, and Assumption 3.12 to hold with $\epsilon = \mathcal{O}(\epsilon_0)$.*

Noticing that,

$$\mathcal{W}_2(\overrightarrow{P}_\delta, \overrightarrow{P}_0) \leq \sqrt{\mathbb{E}|\overrightarrow{X}_\delta - \overrightarrow{X}_0|^2} \leq \sqrt{2d\delta},$$

the complexity bound can also be derived with respect to $\overrightarrow{P}_0$. More precisely, we consider the following practical scenario. Now we assume $R^2 = \mathcal{O}(d)$, then optimizing $\delta$ to achieve $\mathcal{W}_2(\overrightarrow{P}_0, \overleftarrow{Q}_{1-\delta}) = \mathcal{O}(\epsilon_0)$ requires at most logarithmic complexity with $\log N = \mathcal{O}\left(\frac{d^3}{\epsilon_0^4}\right)$.

The conclusion and the discussion of future research directions are provided in Appendix E.

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
