# OpenReview forum: "Pathway to $O(\sqrt{d})$ Complexity bound under Wasserstein metric of flow-based models"
_ICLR.cc/2026/Conference — Submitted to ICLR 2026_

### Official Review · Reviewer_Gq2n · 2025-10-22

**Soundness:** 3
**Presentation:** 3
**Contribution:** 2
**Rating:** 4
**Confidence:** 4

**Summary:**

The paper under consideration provides with theoretical guarantees of convergence for a class of flow models built upon the Föllmer process. The Föllmer process is solution to a linear stochastic differential equation, it interpolates between a Gaussian probability measure and a target distribution in finite time. The generative models considered here are obtained considering the time-reversal of the Föllmer process and mimicking its marginal flow through an ordinary differential equation whose velocity field is given in terms of the score function of the forward flow. The main contribution of this work is to obtain convergence guarantees in Wasserstein distance that imply that the number of iterations required to achieve a given precision  is $O(\sqrt{d})$. The most relevant assumption required is that the data distribution is a perturbation of a Gaussian distribution. More precisely, the authors work under what they call *Gaussian tail assumption* (Ass. 3.7). Other relevant assumptions include Lipschitzianess of the approximated scores and a $L^2$ bound for the difference between the true scores and the approximated one. The main challenge in obtaining these bounds, as in most theoretical works on diffusion models, is to handle the time-discretization error. In order to do this, the authors show that the backward velocity field is globally Lipschitz in space, with a time-dependent constant. Moreover, they also show that time-derivative grows at most linearly in the space variable.

**Strengths:**

The strength of the paper lies in the fact that their complexity bound scales like $O(\sqrt{d}/\varepsilon_0)$, where $\varepsilon_0$ is the sought precision. This behavior is optimal, and thus the result is of interest for the community of researchers interested in the theoretical aspects of score diffusion models and flow models.

**Weaknesses:**

The main problem is in the novelty of the results.

- Theorem 3.5 appears to me to follow from Cauchy-Lipschitz theory and and application of Gronwall's lemma, perhaps in a non-standard context. If I am wrong, I'd be happy to review my statement.
- Theorem 3.8 is theoretically interesting as it proposes two-sided bounds for time and space derivatives of the score function. However, similar bounds have already been obtained before in various settings, which are comparable to the Gaussian tail assumption. Some of these contributions are mentioned in the paper. I also suspect that  given the natural connection between the score and Hamilton-Jacobi equations, recent results on the propagation of convexity for HJB  equations such as [1] could be leveraged to obtain dimension free bounds for $ DV $ in a far more general setting than the Gaussian tail assumptions. For example, [1] shows that these new propagation of convexity results recover as a special case the results of Pedrotti and Brigati, which are comparable to the bounds obtained here.  Theorem 3.8  should be compared to these new findings. This comparison could also help in understanding its novelty. Even if we restrict to the results cited in the paper, while the authors report in the appendix the precise statements proven in concurring works, they don't really make a thorough comparison. For example, what's new in comparison with the paper of Pedrotti and Brigati? What is the role played by the bound on the second derivative of $h$, which they don't assume? If we use the Pedrotti and Brigati bounds in the proof of Theorem 3.15 do we get a complexity bound with poorer behavior with respect to the dimension?

- Theorem 3.8 also features bounds on the time-derivative od $ V $. Are these bounds a consequence of the bounds on $ DV $ ? If not, what are the conceptual difficulties to be overcome to pass from a bound on $ DV $ to a bound on $ \partial_t V $?

- About Theorem 3.15: In the context of classical DDPMs $ O(\sqrt{d} / \epsilon) $ complexity bounds have been achieved in [2] under comparable assumptions, namely that the score is globally Lipschitz. Would your proof methods apply even in this case? The remark on page 8 tells, and I agree with this, that Prob ODE is a time-rescaling of the Follmer flow.  Thus, I am left wondering  why is the analysis of Follmer GMs different than the one carried out for classical DDPMs and why the methods of [2] could not yield comparable results to Theorem 3.15


[1] *Chaintron, L. P., Conforti, G., & Eichinger, K. (2025). Propagation of weak log-concavity along generalised heat flows via Hamilton-Jacobi equations. arXiv preprint arXiv:2508.07931.*

[2] *Arsenyan, V., Vardanyan, E., & Dalalyan, A. (2025). Assessing the Quality of Denoising Diffusion Models in Wasserstein Distance: Noisy Score and Optimal Bounds. arXiv preprint arXiv:2506.09681.*

**Questions:**

- Can you make precise the sentence on page 6 where you say that weak semi log-concavity as in Bruno&Sabanis only achieves $O(d)$ sampling complexity?

---

> ### Author Response · Authors · 2025-11-20
>
> Thank you for the helpful feedback. We have revised the manuscript to clarify the novelty and address your comments. Our detailed responses follow below.
>
> **W1.** We agree with the reviewer that the overall structure of our proof follows the classical Cauchy-Lipschitz theory combined with a Gronwall-type argument. In fact, works such as Boffi et al. explore related ideas within a similar framework, but do not provide assumptions of the same explicit, verifiable form as ours. In our case, we show that the structure (pathway) is valid with Follmer flow under a Gaussian tail condition. The latter one allows us to derive explicit dependence of the Lipschitz coefficient and thereby ensures that the propagation term is integrable. Another key technical contribution is establishing an upper bound for $|\partial_t V(t,x)|$, which we show scales as $O(|x|+\frac{1}{\sqrt{1-t^2}})$-an estimate essential for ensuring integrability and achieving the desired dimensional dependence.
>
> **W2.** We are grateful to the reviewer for drawing our attention to these very recent (concurrent) related works. We attempted to compare our results with Proposition 2.10 of [1], but the presence of $\|\nabla^2 h\|_\infty$ makes a precise comparison difficult. A similar issue arises in Appendix A, Corollary A.9, where we compare with Pedrotti \& Brigati. This term arises from the Hessian bound under the Gaussian-tail assumption. It is necessary to control $|\partial_t V|$, which further allows us to achieve the $O(\sqrt{d})$ sampling complexity (see our detailed response to Comment 3), with nearly optimal estimates of the first derivative of the score. In contrast, the second derivative is handled through corresponding assumptions. Since $|\partial_t V|$ is central to the convergence analysis, we compare our bounds with those in Ding et al.\ and Gao et al.\ (2024). Their analyses avoid the blow-up of $|\partial_t V(t,x)|$ near $t=0,1$ by restricting the time interval or using Lipschitz constants scaling as $O(\delta_0^{-2})$. In Wang\&Wang, they bypass the estimate by a martingale approach due to the stochastic sampling of the diffusion model. In contrast, our method ensures that $| \partial_t V|$ is integrable over the full interval $[0,1]$, enabling training and sampling without early stopping.
>
> We agree that weak log-concavity is more general than the Gaussian-tail assumption. Prior works (e.g., Bruno & Sabanis) only achieve $O(d)$ sampling complexity, while we reach $O(\sqrt{d})$. Reference [1] does not analyze convergence or complexity in this setting. Verifying its results in our framework is challenging due to the required Lipschitz properties. We plan to explore such extensions in future work, as mentioned in Appendix E.

---

> > ### Author Response · Authors · 2025-11-20
> >
> > **W3.** Thank you for the feedback. The bounds on $\partial_t V$ depend on $DV$, as discussed in Theorem 3.8 and Table 3 (Appendix B.2). Establishing these bounds is technically challenging, especially for certain terms in the expanded expression. We address this by imposing boundedness on $\|\nabla^2 h\|_\infty$ and applying the Brascamp-Lieb inequality in the log-concave regime, which helps manage singularities and justifies the need for bounds on the second derivative of $h$. We have added clarification below Theorem 3.8.
> >
> > **W4.** We are grateful to the reviewer for raising this point and would like to use this opportunity to provide clarification.
> >
> > Assumption 1 in [2] imposes a uniform upper bound on the posterior variance $Var(X|X+\sigma\xi=y)$, which ensures that the drift coefficient is gradient-Lipschitz. Though not directly assumed on the target density, our G-tail condition may also satisfy this assumption. Since we were not aware of this work before the review, we have not yet verified this connection, but we will carefully examine it in future work. We appreciate the reviewer for drawing our attention to this valuable concurrent work (first posted on arXiv in Jun 2025), and we will cite [2] in the revised manuscript.
> >
> > While the learning processes ensure the margins of Prob-ODE models and DDPMs are related to reverse-time heat flow, the sampling processes differ. The former, such as the  Follmer flow, yields a deterministic flow whose analysis relies on Lipschitz changes of variables. In contrast, DDPMs are based on the reverse-time SDE of a diffusion process, and their guarantees are consequently derived from stochastic-process arguments rather than Lipschitz transport, so that Lipschitz changes of variables are not directly accounted for.
> >
> > Therefore, the DDPM model described in [2] cannot be directly transferred to our setting, and establishing results comparable to Theorem 3.15 would likely require additional techniques beyond the scope of our current method.
> >
> > **Q1.** We appreciate your question and would like to explain it in detail. In Bruno\&Sabanis's work, there are two theorems establishing convergence and achieving an $O(d)$ sampling complexity: Theorem 19 establishes that the $W_2$ error is bounded by $C_4(T,\epsilon)\gamma^{1/2}$, where $C_4 = O(\sqrt{d})$. This implies a sampling complexity of $N = T/h = O(dT/\epsilon^2)$. Similarly, Theorem 21 gives a $W_2$ error bounded by $\tilde{C}_4(T,\epsilon)\gamma^{\alpha}$ with $\tilde{C}_4=O(d)$ and an optimal convergence rate $\alpha \in [1/2, 1]$. When $\alpha = 1$, this yields $N = T/h = O(dT/\epsilon)$. Therefore, Bruno\&Sabanis only achieves $O(d)$ sampling complexity.
> >
> > We hope the revisions satisfactorily address your concerns and might be reflected in a higher evaluation. If the revisions have not fully met your expectations, we would be glad to continue the discussion during the discussion phase.

---

> ### Comment · Reviewer_Gq2n · 2025-11-25
>
> I thank the authors for addressing my questions. Some points are now clearer, such as the usefulness of bunds on $\nabla^2h$ to control $\partial_t V$. I am not sure about its necessity, though. Concerning the comparison with the paper of Dalalyan et al, I still believe that the results of this work attenuate the novelty of the results the work under consideration here. The results therein essentially require less assumptions (the Gaussian tail assumption implies the bound on conditional covariances needed there), and apply to a model that is not very different from the one considered here.
>
> In view of these considerations, I will slightly upgrade my rating of the paper.

---

> > ### Author Response · Authors · 2025-11-28
> >
> > While we believe the two works the reviewer mentioned shall be viewed as parallel works, we still appreciate the effort the reviewer made during this special period to re-evaluate our submission and improve the grade.

---

### Official Review · Reviewer_oCNt · 2025-10-30

**Soundness:** 4
**Presentation:** 3
**Contribution:** 4
**Rating:** 8
**Confidence:** 3

**Summary:**

The paper establishes a new convergence result for the probability flow model.
Under some regularity assumptions and a G-tail assumption for the decay of the data distribution, it is demonstrated the probability flow with constant step size has a Wasserstein discretization error that scales as $O(\sqrt d)$, where $d$ is the dimension of the data space.
The analysis is based on regularity properties of the velocity fields which induces the Lipschitzness of the Föllmer flow.

**Strengths:**

* The new bound get rid of undesirable log terms present in previous works (see Table 1).
* The mathematical presentation is very rigorous.
* All involved constants are explicit in the Tables given in appendix.
* The analysis is general and does not require adapting the flow sampling scheme.

**Weaknesses:**

The paper suffer for some presentation issues:
* l. 190: The $L_2$ (unusual $\mathbb{L}$ ?) loss still involves the unknown score.
* Is it really useful to consider a Föllmer flow with a generic covariance matrix $C$? Is there machine learning applications with non-identity covariances?
* The main results (eg Corollary 3.16) are stated with $M_0$ without precise mention to the dimension $d$, contrary to the abstract and the introduction (Table 1). I would suggest to highlight that $M_0$ is the order of $d$ for $C=I_d$ just after Assumption 3.1, and possibly provide bounds involving $d$
for Corollary 3.16 and Corollary 3.20.


Minor remarks:
* l. 037: with deterministic flow given initial
* l. 041: While for structured data -> However?
* l. 134: the the
* l. 155: can be interpret as approximation
* l. 215: later in see Theorem 3.8 for the regularity of the velocity field and Lemma 3.10 for the proof of well-posedness
* l. 335 Specify that Table 3 is in Appendix

**Questions:**

line 110 : "we achieve an optimal dependence of O √d on the data dimension d" (also in abstract): Can you please clarify the sense of "optimal" here?

---

> ### Author Response · Authors · 2025-11-20
>
> We thank the reviewer for the insightful feedback. We have thoroughly revised the manuscript based on your comments and believe that the improvements clarify and strengthen the work. We would be glad to continue the discussion during the discussion phase.
>
> **W1.** We greatly appreciate the reviewer for highlighting this point. Based on the forward process defined in (6), the marginal distribution $\bar{p}_t$ has an analytical solution
> $$\bar{p}_t(X,t)
> =\int\_{\mathbb{R}^d}(2\pi t(2-t)C )^{-d/2}
> \exp\left(-\frac{\|X-(1-t)X_0\|^{2}}{2t(2-t)C}\right)p\_{data}(X_0)dX_0.$$
>
> We introduce the notation $X_t$ to denote $(1-t)X_0 + \sqrt{t(2-t)C}\,\mathcal{N}$. Note that although it shares the same marginal distribution as the $\overset{\rightarrow}{X}_t$ in the forward process (6) in the paper, it is different and is introduced solely for notational convenience. According to the result of Yubin\& Wang, the global score function $S$ can be interpreted as a conditional expectation, namely
>
> $$S(1-t,X)=C\nabla \log \bar{p}\_t=\frac{\nabla\bar{p}\_t}{\bar{p}\_t}=\mathbb{E}_{X_0|X_t}\Bigg[-\frac{X_t-(1-t)X_0}{t(2-t)} \bigg|X_t=X\Bigg].$$
>
> Then by definition of velocity field $V$, we have
> $$V(1-t,X):=\frac{1}{1-t}\bigl[X+S(1-t,X)\bigr]=\mathbb{E}_{X_0|X_t}\left[\frac{1}{1-t}X_t-\frac{X_t-(1-t)X_0}{(1-t)t(2-t)}\bigg| X_t=X\right].$$
> Consequently, the $\mathbb{L}\_2$ (in the sense of expectation $\mathbb{E}$) estimation loss (9) is equivalent to seeking a network-represented function $\tilde{V}$ that minimizes
>
> $$\mathbb{E}_{X_0,\; N\sim\mathcal{N}(0,I_d),\; t}\left[\lambda(t)\left\|\tilde{V}\left(1-t,X_t\right)-\frac{1}{1-t}X_t+\frac{\sqrt{C}\mathcal{N}}{(1-t)\sqrt{t(2-t)}}\right\|^{2}\right].$$
>
> To avoid any possible confusion for readers, we have incorporated this clarification into L. 197-208 of the manuscript.
>
> **W2.** We sincerely thank the reviewer for their insightful comments. We wanted to make our statement more general: covers both standard application $C=I_d$ and dimension independent bound when $C$ is trace class. The latter arises in the posterior of some Bayesian inverse problems and has been elaborated upon in reference [1]. Our purpose here is simply to illustrate the applicability of our method and demonstrate its implementation in the Bayesian inverse problem setting, as shown in Section "Convergence in the Bayesian Inverse Problems" in Appendix D.
>
> [1] Jae Hyun Lim, Nikola B. Kovachki et al. Score-based diffusion models in function space. Journal of Machine Learning Research, 26(158):1--62, 2025.
>
> **W3.** We appreciate this helpful suggestion and have added the corresponding clarification in the revised manuscript(Under Assumption 3.1, Theorem 3.5, Theorem 3.15 Corollary 3.16, Corollary 3.20, Corollary 3.16 and Corollary 3.20). We also added narrative throughout the manuscript to emphasize the relationships among the theorems, producing a more structured presentation. We acknowledge that this improves readability for the audience, which has also been raised by multiple reviewers.
>
> **W4.** We thank the reviewer for the comment and have corrected the sentence as suggested.
>
> **Q1.** We thank the reviewer for bringing this to our attention and would like to use this opportunity to offer a clarification. "Optimal" refers to the fact that our approach attains the $O(\sqrt{d})$ scaling, which is the best achievable dependence on the dimension under our assumptions. This can be understood from the following aspects:
>
> * Our method attains the $O(\sqrt{d})$ complexity bound under Gaussian-tail assumptions, improving upon concurrent works such as Bruno \& Sabanis (2025) under weak log-concavity assumptions, which only achieve $O(d)$ scaling (See the first paragraph after Corollary 3.16).
>
> * Proposition 6 in Gao (2025) shows that, for the *standard Gaussian target distribution*, the $O(\sqrt{d})$ complexity bound is indeed optimal.  This indicates that our $O(\sqrt{d})$ dependence stems from intrinsic Gaussian concentration, making the dimensional scaling fundamental rather than algorithm-induced. In addition, we also provide a rigorous proof framework. Although we do not provide a lower bound on the complexity, we believe that our result represents the best achievable rate under the current framework (See the first paragraph after Corollary 3.16).
>
> * Under our Gaussian-tail assumption, the Föllmer flow is regular at both $t=0$ *and* $t=1$, which allows convergence over the entire interval $t\in[0,1]$ without truncation or restrictive stability conditions (in contrast with Ding et al. (2023) and Gao et al. (2024)) (See the first paragraph after Theorem 3.8). This full-interval regularity further enables us to remove the undesirable logarithmic factors present in previous analyses Wang \& Wang (2024) with
> $N = O\left(\frac{M_0}{\epsilon_0^2}\left(\log\frac{M_2 + \mathrm{Tr}(C)}{\epsilon_0^2}\right)^3\right)$ (See Appendix C).

---

### Official Review · Reviewer_fxPs · 2025-10-31

**Soundness:** 2
**Presentation:** 1
**Contribution:** 2
**Rating:** 2
**Confidence:** 3

**Summary:**

This paper conducts theoretical analysis on the Wasserstein based error of Follmer Process based on the discretization step of the Euler approximation and the dimension of the dataset. The bound is provided based on various regularity assumption on the flow as well as the accuracy of the arpproximation, as well as the smoothness assumption on the dataset distribution that generalizes the semi-log-concavity assumption.

**Strengths:**

The paper is well presented upto some point, and the problem itself is well stated and the assumed regularity assumptions on the flow are fair.  The paper presents the lower complexity bound than previous works for a Target with more general conditions than the previous works.

**Weaknesses:**

The paper is hard to follow for many reasons.

- The claimed statements are not directly referenced in the main results. For example, I believe that the main contribution of this paper is 3.15, and it is stated that this is the result that proves $\sqrt{d}$ bound.  But there in no direct appearance of $d$ in the statement of itself. The author is most likely refering to $M_0$, which is defined as $max (Tr(C)), M_2)$, and  that for the isotropic $C$ in the assumption,  $Tr(C) \sim d$ so that $M_0 \in O(d)$.   Such reasonings are implicitly assumed and used throughout the statements, an the reader is required to go back and forth between the Constant Table4 and the stated theorems/corollaries to check the validity of the claim.  I strongly believe this is not a reader's job to disect the implicit statement on the results that is being claimed "main".

- The paper claims, in the abstract, that " the error can be explicitly controlled by two parts: the Lipschitzness of the push-forward maps of the backward flow which scales independently of the dimension; and a local discretization error scales with $\sqrt{d}$".  But these (2) parts seems not be explicitly described in the work.  I can infer that the first part is corresponding to $\exp$ part that rises naturally from the time-propagating discrepancy of the semigroup maps, and the latter part is corresponding to the part that is related to $M_0$, which is probably related to $d$ via the Assumption 3.1.
I believe that when such a statement is made in abstract, it must be referenced in the writing to clarify the author's intention.

- The claim in 3.20 seems very ambiguous and possibly misleading. The same applies to 3.16 as well. It is being claimed that this oder of $N$ is a "requirement". I believe taht this bound is derived from 3.19, but if it is a "requirement", do we not need the tightness of the bound? $W_2(P, Q)^2  \leq E [|X - Y|^2] $ is not tight as well in the case of this paper---the Follmer flow is not an Optimal Transport.

- The paper ends abruptly without leaving any remarks for the main claims made in the paper,  is this really the final version of the submission?  I feel that this is a paper with the right idea but it is in a format not fit to the venue.  This paper's worth itself is noted, but major revision seems necessary.

**Questions:**

Please see the Weakness section.

Also, what is the proof technique on this paper differ from the previous works that allows the tighter complexity bound? Is there any distinguishing regularity assumptions in the Follmer flow that is not used in the previous works?  ( G-tail seems like more general assumption, so it must be making the proof harder.)

**Details Of Ethics Concerns:**

This is a theory paper.

---

> ### Author Response · Authors · 2025-11-20
>
> We thank the reviewer for the insightful feedback. We have thoroughly revised the manuscript based on your comments and believe that the improvements clarify and strengthen the work. Below we list the detailed reply.
>
> **W1.** We would like to thank the reviewer for their careful reading of our manuscript and for their helpful suggestions. We agree that the original description may appear ambiguous. We intended to present a general solution framework that encompasses both the finite-dimensional case and its extension to infinite-dimensional settings. We acknowledge that this may not have been sufficiently explicit in the original version. To improve readability and better convey this intended generality, we have followed the reviewer's suggestion and added clarifications in the revised manuscript, explicitly specifying that $M_0 = \mathcal{O}(d)$ in the relevant parts (Under Assumption 3.1, Theorem 3.5, and Theorem 3.15).
>
> Theorems 3.8 and 3.15 involve a substantial number of coefficients, and presenting them inline would significantly interrupt the flow of the main exposition. To maintain readability, we summarize these coefficients in tables 3 and 4. Due to page limitations, we were compelled to place these tables in the appendix, which may require some back-and-forth reading. We note, however, that all coefficients $K_1$-$K_9$ are dimension-free, and we have made this explicit in the revised manuscript.
>
> **W2.** We sincerely thank the reviewer for the insightful comment. As clarified after Theorem 3.5, the two parts mentioned in the abstract correspond to: (1) the term $\left(\prod_{j=0}^{N-1}\text{Lip}(\tilde{T}\_j)\right)\mathcal{W}\_2(\overset{\leftarrow}P\_{0}, \overset{\leftarrow}Q\_{0})$ , which captures the effect of the Lipschitzness of the push-forward maps of the backward flow, and (2) the term $\left(\Big(\bar{K}\sqrt{M_0}+\frac{\bar{K_1}}{\sqrt{1-t_j^2}}+\bar{K_2}\Big)h+\epsilon\right)$ which represents the accumulated local discretization error, scaling as $\sqrt{M_0} \in \mathcal{O}(\sqrt{d})$, highlighting the square-root dependence. We have addressed this point in the revised version (under Theorem 3.5 and Theorem 3.15), where we have added additional discussion and clarification. In addition, we have added explanatory text throughout the manuscript to better connect the individual theorems.
>
> **W3.** We sincerely thank the reviewer for the valuable comments and suggestions. We agree that the use of "require" is not appropriate here. We intend to show a pathway to an $\mathcal{O}(\sqrt{d})$ complexity bound for flow-based models under the Wasserstein metric, and in particular to provide an **upper bound** on the complexity needed to achieve $\mathcal{W}_2(\overset{\rightarrow}{P}\_0, \overset{\leftarrow}{Q}\_{1}) = \mathcal{O}(\varepsilon_0)$ with uniform step sizes, rather than a lower bound. By referring to Ref.~[1], we have therefore revised the wording from "require" to "require at most" in the updated manuscript.
>
> **W4.** We greatly appreciate the reviewer for highlighting this point and would like to take this opportunity to clarify. Our presentation was intentionally concise due to strict page limitations, which required us to prioritize the core contributions of the paper—namely, identifying a viable proof pathway (Theorem 3.5) and demonstrating its validity through the Föllmer flow under Gaussian tails. We appreciate the reviewer's suggestion and have added a dedicated section *Conclusion and Future Directions* in the revised manuscript, including relaxing the Gaussian tail assumption, advancing higher-order discretization theory, clarifying training-related effects, optimizing step sizes, and developing data-driven Lipschitz estimates--all of which could further strengthen the theoretical and practical guarantees of flow-based models. Due to page constraints, this section is included in Appendix E.

---

> > ### Author Response · Authors · 2025-11-20
> >
> > **Q1.** We sincerely thank the reviewer for their insightful question. As mentioned in our response to Comment 2, the total error is controlled by two parts, among which the proof of the second term - local discretization error scaling with $\sqrt{d}$ poses the main challenge and the proof of it is technical.
> >
> > The *key proof technique* appears in the first paragraph following Assumption 3.7: by leveraging the Gaussian-tail assumption to control the regularity of the velocity field, we obtain a tighter complexity estimate than previous works. This enables us to derive an explicit upper bound on $|\partial_t V(t,x)|$ using tools such as the Brascamp–Lieb inequality, which had not been utilized in prior analyses. Compared with Ding et al. (2023) and Gao et al. (2024), we further show that the factor is $O(\frac{1}{\sqrt{1-t^2}})$, which is integrable over $[0,1]$ and ensures convergence across the whole interval $t \in [0,1]$. To the best of our knowledge, our work is the first to adopt this approach and to show that it yields the improved $\mathcal{O}(\sqrt{d})$ complexity bound.
> >
> >  We hope the revisions satisfactorily address your concerns and might be reflected in a higher evaluation. If the revisions have not fully met your expectations, we would be glad to continue the discussion during the discussion phase.
> >
> > [1]Joe Benton, Valentin De Bortoli, Arnaud Doucet, and George Deligiannidis. Nearly $d$-linear convergence bounds for diffusion models via stochastic localization. URL https://openreview.net/forum?id=r5njV3BsuD

---

### Official Review · Reviewer_Q1jr · 2025-10-31

**Soundness:** 3
**Presentation:** 1
**Contribution:** 2
**Rating:** 2
**Confidence:** 3

**Summary:**

This work analyzes accumulated error for Euler step sampling using Follmer flows. The main contribution is a tighter bound in terms of data dimension $d$ along with the analytical tools.

**Strengths:**

A tighter bound in terms of $d$ is provided.

**Weaknesses:**

1. The entire paper seems rushed and incomplete. Section 3 seems to be a compilation of assumptions, corollaries and theorems. Interpretations and explanations for stated results are barely provided. There is no conclusion & future work section. Overall, I don't think the current presentation helps the audience to connect the assumptions and bounds to the machine learning context. I'd advise the authors to present a simplified set of assumptions and theorems in the main paper to have more space to convey the intuition and implications of these results while deferring the formal, rigorous presentation in the appendix.

2. As a consequence of 1, I'm not able to understand the significance of this work.

3. Follmer flows seems to be a special case of rectified flow [1] (see appendix A in [2]) by letting the noisy prior distribution to be $\mathcal{N}(0, C)$. Mathematically, Follmer flow seems to be $X_t  = t X_1 + (1-t)X_0$ but restricting $X_0$ to be sampled from $\mathcal{N}(0, C)$.  Can the authors confirm this observation? If so, I think it would be necessary to further discuss the connection between this work and rectified flows.


4. I highly doubt that assumption 3.13 would hold in practice. It basically requires a neural network to be uniformly bounded on a non-compact support.


5. Would it be easier to bound/spell out $Lip(T_j)$ and $Lip(\tilde{T}_j)$ in terms of quantities related to the score function/ neural networks? It would help the audience to connect the assumptions back to properties of neural nets.

[1] Liu, X., Gong, C., & Liu, Q. (2022). Flow straight and fast: Learning to generate and transfer data with rectified flow. arXiv. https://arxiv.org/abs/2209.03003

[2] Rout, L., Chen, Y., Ruiz, N., Caramanis, C., Shakkottai, S., & Chu, W.-S. (2024). Semantic image inversion and editing using rectified stochastic differential equations. arXiv. https://arxiv.org/abs/2410.10792

**Questions:**

1. what is the significance to analyze a general $C$ instead of $C=I$? I've only seen the use of correlated noise in infinite dimensional flows like [1] and [2] only because white Gaussian noise is not trace-class in a general Hilbert space. But it's certainly not the case here.

2. see weakness.


[1] Kerrigan, G., Migliorini, G., & Smyth, P. (2023). Functional Flow Matching. arXiv. https://arxiv.org/abs/2305.17209

[2]Franzese, G., Corallo, G., Rossi, S., Heinonen, M., Filippone, M., & Michiardi, P. (2023). Continuous-Time Functional Diffusion Processes. arXiv. https://arxiv.org/abs/2303.00800

---

> ### Author Response · Authors · 2025-11-20
>
> Thank you for the helpful feedback. We have revised the manuscript to clarify the novelty and address your comments. Our detailed responses follow below.
>
>
> **W1**\&**W2.** We sincerely thank the reviewer for the insightful comment. The significance of our work lies in developing analytical tools to bound error accumulation in high-dimensional sampling flows under the Wasserstein metric, thereby ensuring optimal iteration complexity. These results are directly relevant to machine learning tasks such as generative modeling and Bayesian inference, including infinite-dimensional settings; In Sec. 3.1, we develop a general flow-based framework, not limited to the Föllmer flow. Sections 3.2-3.3 validate the assumptions and establish complexity results under the Gaussian tail condition, Sec. 3.4 addresses low-dimensional manifold structures in real-world data. Due to page limitations, we have moved the discussion of the relation to probabilistic ODEs and Bayesian inverse problems to Appendix C and Appendix D. We have also added connecting and explanatory text between Sections 2 and 3, as well as a dedicated section, Conclusion and Future Directions, in the revised manuscript (included in Appendix E), which outlines several promising directions such as relaxing the Gaussian tail assumption, developing higher-order discretizations, and examining the effects of different training objectives, among others.
>
> **W3**.  We greatly appreciate the reviewer for highlighting this point and would like to take this opportunity to clarify. As the name suggests, the rectified flow seeks straight line coupling of input and generation through rectification. While the Föllmer flow in general follows the heat flow.
> The Föllmer flow's velocity feild $V(x,t)$ satisfies a viscous Hamilton-Jacobi PDE with both quadratic gradient and Laplacian terms,
>
> \begin{equation*}
> \partial_t V + (1-t)\left(2 (\nabla V) V- \nabla( \nabla\cdot V)\right)= 0.
> \end{equation*}
> In contrast,  the velocity of rectified flows $V$ is constant along straight-line trajectories,
> \begin{equation*}
> \frac{d}{dt}V(x_t,t) = \partial_t V + (\nabla V)V = 0,
> \end{equation*}
> without Laplacian terms which leads to the non-viscous Hamilton-Jacobi PDE. Therefore, the connection between the Föllmer flow and the rectified flow is not direct.
>
> We mention rectified flow in the manuscript because it's an important case within ODE-based generative model. In the revision, we have removed the explicit expression $X_t = tX_1 + (1-t)X_0$ to avoid notational confusion, while also conserving space in the paper.
>
> **W4** We thank the reviewer for the valuable feedback. We agree that the constant $K_8$ in Assumption 3.13 may not be small without further assumptions. While we believe the assumption remains reasonable for the following reasons.
> - The true velocity field $\nabla V(t,x)$ is analytically verified to satisfy the required growth condition, and since $\widetilde{V}(t_n,x)$ is trained to approximate $V(t_n,x)$, it is natural to assume a similar bound with an additional constant $K_8$.
> - The neural networks typically inherit regularity and controlled growth from their architecture and training, making unbounded behavior unlikely in practice. For instance, a ReLU neural network, after being trained to approximate $V$, becomes a piecewise linear function with a finite partition; the derivatives are hence bounded in terms of weight and bias. We have added the corresponding explanation under the assumption 3.13 in the manuscript.
>
> We also add Remark B.1 in the revised manuscript: Assumption 3.13 can be relaxed in the temporal $t$ direction: it suffices to require that the discrete-time accumulated contribution of the learned velocity field is bounded.

---

> > ### Author Response · Authors · 2025-11-20
> >
> > **W5.** We sincerely thank the reviewer for the insightful comment. Recall Eq. (8) that the velocity field is
> > \begin{equation*}
> > V(t, x) = \frac{1}{t}\left[x + S(t,x)\right], \text{with score function $S(t,x)=C\nabla \log p_{t}(x)$}.
> > \end{equation*}
> > Theorem 3.8 provides the bound on $\lVert \nabla V(t,\cdot)\rVert_\infty$, and Assumption 3.13 imposes a similar restriction on the network approximation $\lVert \nabla \tilde V(t,\cdot)\rVert_\infty$. Using these estimates, the proofs of Corollaries 3.11 and 3.14 yield bounds on the corresponding Lipschitz constants. In particular, in the proof of Theorem 3.15 (see L .1475-1481 in Appendix B.6), we explicitly write the relationship according to Proposition A.7 in Mikulincer \& Shenfeld(2023):
> >
> > $$
> > \text{Lip}(T_n)
> > \le
> > \exp\\left(
> > \int_{t_n}^{t_{n+1}}
> > \lVert \nabla V(t,\cdot)\rVert_\infty, dt
> > \right),\quad\text{Lip}(\overset{\leftarrow}{X}_1(x)) \leq \prod _{j=0} ^{N-1} \text{Lip}({T}_j),
> > $$
> > and
> >
> > $$
> > \text{Lip}( \\tilde T_n)
> > \le
> > 1+(t_{n+1}-t_n) \lVert \nabla \tilde V(t,\cdot)\rVert_\infty, \quad \text{Lip}(\overset{\leftarrow}{Y}_1(x)) \leq\prod _{j=0} ^{N-1} \text{Lip}(\widetilde{T}_j).
> > $$
> > These bounds explicitly connect the Lipschitz coefficient to properties of the score function and the neural network.  To enhance readability, we highlight the connection between $\mathrm{Lip}(T_n)$ ($\mathrm{Lip}(\widetilde{T}_n)$) and the  score function (neural networks)-directly below Assumption 3.2 and Assumption 3.3.
> >
> > **Q1.** We are grateful to the reviewer for the thoughtful feedback. We consider a general covariance matrix $C$  to cover both the identity case $C = I_d$ and the correlated case $C \neq I_d$. In the main text, we primarily focus on the former, which is interpreted as a generative model on a homogeneous $d$-dimensional subspace, for instance, text and image generations. Under such a setting, $\operatorname{Tr}(C)=d$, which implies $M_0=\mathcal{O}(d)$, reflecting the growth of second-order energy with the ambient dimension. We agree with the reviewer that the use of correlated noise (i.e., $C \neq I_d$) is mainly relevant in infinite-dimensional flows, and we discuss this case in Appendix D in the context of Bayesian inverse problems.  To improve readability and better convey this intended generality, we have added clarifications in the revised manuscript, explicitly specifying that $M_0 = \mathcal{O}(d)$ in the relevant parts (Under Assumption 3.1, Theorem 3.5 and Theorem 3.15).
> >
> > We hope the revisions satisfactorily address your concerns and might be reflected in a higher evaluation. If the revisions have not fully met your expectations, we would be glad to continue the discussion during the discussion phase.

---

> ### Comment · Reviewer_Q1jr · 2025-11-21
>
> Thanks for the response. For W3, if you compare SDE in (21) in [1] against (5) in this work and (23) in [1] against the definition of V in (8) in this work, I think they clearly resemble, up to the point that [1] considers a standard Gaussian as a reference distribution and this work considers a gaussian with a non-identity correlation C. The close resemblance does not seem to be coincidental, and " connection between the Föllmer flow and the rectified flow is not direct" seems unconvincing by comparing the maths in this paper and [1]. Can the authors please clarify?
>
> [1] Rout, L., Chen, Y., Ruiz, N., Caramanis, C., Shakkottai, S., & Chu, W.-S. (2024). Semantic image inversion and editing using rectified stochastic differential equations. arXiv. https://arxiv.org/abs/2410.10792

---

> > ### Author Response · Authors · 2025-11-21
> >
> > We appreciate your question and would like to explain it in detail. First, we would like to clarify that our mentioned “rectified flow" in the paper refers to the work introduced by Liu et al. [2], hence we did not initially examine the forward process used in [1]. We appreciate the reviewer for pointing this out, which allowed us to clearly distinguish the rectified flow (RF) from the Föllmer flow in our work.
> >
> > Although Flow-based models are always fitting marginals of forward process, the **forward SDE** in Eq. (21) of RF **is not equivalent** to Eq. (5) in our paper, even when the correlation matrix $C$ is set to the identity. For clarify, the  forward SDE in  RF is
> > \begin{equation}
> > dY_t = -\frac{1}{1-t}Y_t dt + \sqrt{\frac{\textbf{2t}}{1-t}}dW_t, \quad Y_0 \sim P_0, \tag{21}
> > \end{equation}
> > with the variance term $\frac{\textbf{2t}}{1-t}$. In contrast, our Eq. (5) is
> > \begin{equation}
> > d\overset{\rightarrow}{X}_t = -\frac{1}{1-t}\overset{\rightarrow}{X}_t dt + \sqrt{\frac{\textbf{2}}{1-t}}dW_t, \,\overset{\rightarrow}{X}_0 \sim \nu,\tag{5}
> > \end{equation}
> > with the variance term $\frac{\textbf{2}}{1-t}$. As explained in Jiao et al. [3, Appendix D], Eq. (5) is obtained directly from the **Ornstein-Uhlenbeck** process in Eq. (13) of [1] via change of time formulation $t = 1 -e^{-s}$, a connection also noted by Eric et al.[4, Sec. 5.1]. This difference is fully consistent with the statement in the last paragraph on Page. 6 of [1]:   "our approach is based on rectified flows, which leads to a different ODE and consequently translates into a **different SDE**." This also explains why the velocity field of RF in Eq. (23) of [1] is fundamentally different from the definition of $V$ in Eq. (8) of our work.
> >
> > We further note that the **marginal** induced by the two processes are also **distinct**. As shown below Eq. (9) in our revised manuscript,  the forward process is simplified by $X_t := (1-t) X_0 + \sqrt{t(2-t)}\mathcal{N}$,
> > whereas RF adopts the linear interpolation
> > $Y_t = (1-t) Y_0+t Y_1, Y_1 \sim \mathcal{N}(0,I)$ in [1],
> > which is specific to its rectified-flow design.
> >
> > Taken together, rectified flow and our Föllmer flow are inherently **different**. We will further investigate the regularity properties of RF as a direction for future work.
> >
> > [1] Rout, L., Chen, Y., Ruiz, N., Caramanis, C., Shakkottai, S., \& Chu, W.-S. (2024). Semantic image inversion and editing using rectified stochastic differential equations. arXiv. https://arxiv.org/abs/2410.10792.
> >
> > [2]Xingchao Liu, Chengyue Gong, and qiang liu. Flow straight and fast: Learning to generate and transfer data with rectified flow. In The Eleventh International Conference on Learning Representations, 2023. URL https://openreview.net/forum?id=XVjTT1nw5z.
> >
> > [3]Yin Dai, Yuan Gao, Jian Huang, Yuling Jiao, Lican Kang, and Jin Liu. Lipschitz transport maps via the Föllmer flow. arXiv preprint arXiv:2309.03490, 2023.
> >
> > [4]Michael S Albergo, Nicholas M Boffi, and Eric Vanden-Eijnden. Stochastic interpolants: A unifying framework for flows and diffusions. arXiv preprint arXiv:2303.08797, 2023.

---

> > ### Author Response · Authors · 2025-11-28
> >
> > While awaiting your response, we went a bit deeper into the question you raised.
> >
> > Many existing works show that the flow-based models, including rectified flow and Föllmer flow, are induced by two steps: 1) define some coupling of data distribution $X_1$ and base distribution $X_0$; 2)
> > choosing a general interpolation path of the form $X_t = \alpha_t X_1 + \beta_t X_0$ and fitting the associated velocity field via the so-called Markovian
> > projection [1,3]. Under this unified viewpoint, both the finite-dimensional
> > rectified flow [1,2] and the probability-flow
> > ODE derived from the Föllmer flow (after an appropriate change of time
> > variable) appear as special cases corresponding to particular choices of
> > $(\alpha_t,\beta_t)$.
> >
> > The first case-standard rectified flow-begin with a first step rectification
> > constructed using an independent coupling between $X_0$ and $X_1$. In the
> > subsequent iterative rectification procedure, each iteration builds a new,
> > refined coupling between $X_0$ and $X_1$ from the prior steps with less transport cost. This iterative process admits a
> > stationary point at which the optimal transport map is recovered, and hence we note it as displacement interpolation.
> > By contrast, the second case-the Föllmer flow, whose forward SDE follows the
> > heat flow-is not a straight or displacement-type interpolation. More explicitly, the variance-preserving structure of the heat flow leads to an evolution marginal that differs markedly from that of rectified flows.
> >
> > | Flow path | $\alpha_t$ | $\beta_t$ |
> > |--------|--------|--------|
> > | Displacement interpolation | $t$ | $1-t$ |
> > | Variance preserving interpolation | $t$ | $\sqrt{1-t^2}$  |
> >
> > These observations underscore that despite sharing the common Markovian
> > projection framework, the flow paths generated by standard rectified flow (with either one-step rectification or multistep rectification) and those by the
> > Föllmer flows are  different and should be regarded as distinct
> > constructions.
> >
> > With this observation and our main goal of the manuscript (general pathway to $\sqrt{d}$ complexity bounds of flow-based models), we subsequently validated the feasibility of generalizing our method to the one-step rectified flow setting.
> > In particular, our approach remains applicable to the interpolation paths used in the flow built by the first step rectification over independent coupling in [1,2] prior to the recursive construction, and it achieves a similar $\mathcal{O}(\sqrt{d})$ complexity as stated in Theorem 3.15 on Page 9. The detailed proof is provided below.

---

> > > ### Author Response · Authors · 2025-11-28
> > >
> > > As noted above, the forward process in the first-step rectification, constructed
> > > by independently coupling the data with a standard Gaussian reference
> > > distribution, is defined by the interpolation $\hat{X}_t =\hat{\alpha}_t \hat{X}_1 + \hat{\beta}_t \mathcal{N}$ with $\hat{\alpha}_t = 1-t, \hat{\beta}_t = t$.
> > > Then the transition probability distribution from $\hat{X}_0$ to $\hat{X}_t$ is given by
> > > \begin{equation}
> > > \hat{X}_t |\hat{X}_1 = x_1 \sim \mathcal{N}\big((1 - t)x_1, tI_d\big).
> > > \end{equation}
> > > Under the Gaussian tail Assumption 3.7, the score function can be calculated by
> > > \begin{equation}
> > > \hat{S}(t, x):=\nabla\log \hat{p}_t
> > > =\nabla \log \int _{\mathbb{R}^d} \left( 2\pi\, \text{det} (\hat{B}(t)) \right)^{-\frac{d}{2}}G(t,x,y)\cdot\exp\left(-\frac{|x| _{\hat{A} _{t}}}{2}\right)\exp \left( h(y) \right) \mathrm{d}y,
> > > \end{equation}
> > > where $G(t,x,y):=\exp\left(-\frac{|\hat{K}(t)x -y|^2 _{\hat{B}(t)}}{2}\right)$, $\hat{K}(t)=(A\hat{A} _{t}^{-1})t$, $\hat{B}(t)=(A\hat{A} _{t}^{-1})(1-t)^2$.
> > > First, we consider the modified score function:
> > > \begin{equation}
> > > \widetilde{\hat{S}}(t,x):=\hat{S}(t,x)+\hat{A} _{t}^{-1}x\leq |\hat{K}(t)\nabla h(x)|.
> > > \end{equation}
> > > Let $\hat{K}=\sup _{0\leq t\leq 1}{|\frac{1}{t}\hat{K}(t)|}=\sup _{0\leq t\leq 1}{|A \hat{A} _{t}^{-1}|}\leq1+ \|A\|$, we have
> > > \begin{equation*}
> > > |\widetilde{\hat{S}}(t,\cdot)| _\infty\leq \hat{K}|\nabla h| _\infty t= \hat{K} _0t
> > > \end{equation*}
> > > with $\hat{K}_0:=\hat{K}|\nabla h| _\infty$.
> > > Taking the derivative twice along that direction and using the same method as above, we get
> > > \begin{equation*}
> > > \|\nabla\widetilde{\hat{S}}(t,\cdot)\| _\infty \leq \hat{K}(t)^2(\|\nabla ^2 h\| _\infty+|\nabla h|^2 _\infty)=\hat{K} _1t^2,
> > > \end{equation*}
> > > where  $\hat{K}_1:=\hat{K}^2(\|\nabla ^2 h\| _\infty+|\nabla h|^2 _\infty)$.
> > >
> > > Define $\hat{K} _2:=\sup _{0\leq t\leq1}\|\frac{1}{t}\big(I _d-(1-t)\hat{A} _{t}^{-1})\big\|=\sup _{0\leq t\leq1}\|(A+C-I _d)\big(At^2+(1-t)^2I _d\big)^{-1}\|$, we have
> > > \begin{equation*}
> > > \begin{aligned}
> > >     |\hat{V}(t,x)|:=\left|\frac{x+(1-t)\hat{S}(t,\cdot)}{t}\right|=\left|\frac{(1-t)\widetilde{\hat{S}}(t,\cdot)+\big(I _d-(1-t)\hat{A} _{t}^{-1}\big)x}{t}\right|\leq \hat{K}_0+\hat{K}_2|x|,
> > > \end{aligned}
> > > \end{equation*}
> > > and
> > > \begin{equation*}
> > > \begin{aligned}
> > >     \|\nabla \hat{V}(t,x)\| _\infty=\left\|\nabla\left(\frac{(1-t)\widetilde{\hat{S}}(t,\cdot)+(I _d-(1-t)\hat{A} _{t}^{-1})x}{t}\right)\right\| _\infty\leq \hat{K} _1t+\hat{K} _2.
> > > \end{aligned}
> > > \end{equation*}
> > > Similar to the proof of $|\partial_t V|$ of F\"{o}llmer flow in Appendix B.2, we obtain
> > > \begin{equation*}
> > > \begin{aligned}
> > > |\partial _t\hat{V}(t,x)|\leq  \hat{K} _5|x|+\frac{\hat{K} _6}{\sqrt{1-t^2}}+ \hat{K} _7.
> > > \end{aligned}
> > > \end{equation*}
> > > where  $\hat{K} _5,\hat{K}_6$ and $\hat{K} _7$ are dimension-free constants.
> > > This completes the argument that the trajectory $\hat{X} _t$ of one-step rectified flow possesses the similar regularity properties for its velocity field as those established in Theorem 3.8. Consequently, by following the proof
> > > steps of Theorem 3.15 in Appendix B.6, the desired result directly follows.
> > > This successful incorporation further demonstrates both the generality of our
> > > approach and the broad applicability of the Gaussian-tail assumption.
> > >
> > > We hope this further comment addresses your concern in a more correct way, and we look forward to your positive feedback and further comments.
> > >
> > > [1] Xingchao Liu, Chengyue Gong, and qiang liu. Flow straight and fast: Learning to generate and transfer data with rectified flow. In The Eleventh International Conference on Learning Representations, 2023. URL https://openreview.net/forum?id=XVjTT1nw5z.
> > >
> > > [2] Rout, L., Chen, Y., Ruiz, N., Caramanis, C., Shakkottai, S.,  Chu, W.-S. (2024). Semantic image inversion and editing using rectified stochastic differential equations. arXiv. https://arxiv.org/abs/2410.10792.
> > >
> > > [3] Shi, Y., De Bortoli, V., Campbell, A., Doucet, A. (2023). Diffusion schrödinger bridge matching. Advances in Neural Information Processing Systems, 36, 62183-62223.

---

### Author Response · Authors · 2025-12-02

**Author Summary for AC**

We sincerely thank the AC for their effort in reviewing our submission. Below, we briefly summarize the reviewer-author interactions and score updates, following our detailed responses and substantial revisions to the manuscript.

- Reviewer oCNt assigned an initial rating of 8 and emphasized that our work “provides a rigorous analysis with explicit improved bounds that remove the undesirable log terms of prior work, all without requiring any modification to the flow sampling scheme,” recognizing the novelty of our work and its mathematically rigorous presentation. The reviewer also offered constructive suggestions, such as "clarifying whether the $L_2$ loss still involves the unknown score" and addressing "several minor remarks". We resolved these issues by explaining that "the global score function can be interpreted as a conditional expectation" and we incorporated the corresponding clarifications into the manuscript. After carefully addressing all comments and revising the manuscript, the reviewer maintained the original score, and we sincerely appreciate this positive assessment.

- Reviewer Gq2n initially assigned a rating of 4, noting that “the strength of the paper lies in the fact that their complexity bound scales like $\mathcal{O}(\sqrt{d})$, which is optimal, and thus the result is of interest for researchers working on the theoretical aspects of score diffusion models and flow models.” The reviewer’s main concerns centered on the comparison with concurrent works and several technical aspects of our theoretical proof. To address these issues, we conducted a careful comparison with the concurrent literature and clarified that, although our assumptions are slightly stronger, they are essential for establishing a valid pathway to the convergence result-in particular, the need to control $|\partial_t V|$, which is crucial for achieving the optimal sampling complexity. We also acknowledge that extending our method to weaker conditions, such as weak log-concavity, is an interesting direction for future work, as stated explicitly in the conclusion. After carefully evaluating our rebuttal and the corresponding updates, the reviewer **raised the rating to 6** on 26 Nov 2025, acknowledging the value of our work.

- Reviewer Q1jr assigned an initial rating of 2, and acknowledged that "the main contribution of our work is a tighter bound in terms of the data dimension $d$, along with the analytical tools." However, the reviewer considered the weaknesses to lie mainly in the presentation and interpretation of our main results. We clarified that our central innovation is the development of a unified and implementable **pathway**  that establishes **the optimal**  **$\mathcal{O}(\sqrt{d})$** complexity bound under the Wasserstein metric for flow-based generative models. This pathway may have been envisioned in earlier literature; however, to the best of our knowledge, no prior work has succeeded in turning this idea into a fully rigorous and generalizable proof. Due to the **page limits** of the initial submission, several key motivations are not fully articulated. Given ICLR's broad audience, we have expanded the exposition in the revised manuscript to present this pathway and its implications more clearly and transparently.
We appreciate that the reviewer raised a follow-up question on 21 Nov 2025, and we promptly provided a detailed response on the same day, addressing the new concerns. Since then, we have not received any further reply, although we continued to submit an additional clarifying response and incorporated further discussion of these points into the revised manuscript. We humbly hope that the improvements made during the rebuttal stage and the strengthened manuscript might justify a higher overall evaluation upon the AC’s final review.

---

> ### Author Response · Authors · 2025-12-02
>
> - Reviewer fxPs assigned an initial rating of 2, commenting that “the problem is well stated and the regularity assumptions on the flow are reasonable,” and further noting that "the paper achieves a lower complexity bound under more general conditions than previous works". The reviewer also raised concerns similar to those of Reviewer Q1jr and pointed out that “the relationship between $M_0$ and the $\mathcal{O}(d)$ scaling was not clearly stated.” In fact, our results are formulated for a general covariance matrix $C$, covering both the isotropic case $C = I_d$ and correlated settings $C \neq I_d$. Our main focus is on the isotropic case, where $\operatorname{Tr}(C)=d$, which directly implies $M_0=\mathcal{O}(d)$. The reviewer’s comments further made us realize that some reductions made earlier due to space limitations may have caused ambiguity for readers. We therefore carefully addressed these points and incorporated all corresponding revisions-for example, we added explicit statements in Assumption 3.1, Theorem 3.5, and Theorem 3.15 to make this dependence fully transparent. In addition, the reviewer provided a constructive suggestion regarding “the terminology in Theorem 3.15”, namely that the phrasing should be adjusted to clarify that the result provides an upper bound (“at most”), rather than a lower bound. We have responded positively to this feedback and made the corresponding revision in the manuscript. However, we have not received any further discussion from the reviewer since 21 Nov 2025. We believe that the revised manuscript has significantly improved in completeness and readability, and that the remaining concerns have been fully resolved. We respectfully hope that, after reviewing our detailed rebuttal and the substantially revised manuscript, the AC may consider whether an upward adjustment of the final evaluation is warranted.
>
> We again thank all reviewers for their thoughtful and constructive feedback. We hope that the substantial clarifications and revisions incorporated during the rebuttal stage will be positively reflected in the AC’s final decision.

---

### Meta-Review · Area_Chair_URtK · 2025-12-25

**Summary:**

Reviewers raised concerns about the readability of the paper, about the strength of the assumptions, and the novelty. The authors have put in effort to rewrite parts of the paper, but overall, the scores remain borderline at best.

The crux of the issue, therefore, is the concern regarding novelty. It seems to me that there are two main ingredients in this paper: (1) derivative estimates for the velocity fields, given assumptions on the target distribution (perturbation of a Gaussian; bounded support + early stopping), and (2) ODE discretization analysis. As pointed out by Reviewer Gq2n, ingredient (1) is not new and appears in various papers on heat flow estimates. Here, I would like to point out that bounds on $\partial_t V$ are also not new and have appeared in the diffusion model literature under the name of "score perturbation bounds", see Chen et al. (2023b) and [1]. Arguably, ingredient (2) is also not new: from my understanding, given the Lipschitz assumptions, the discretization bound of $\sqrt d/\epsilon_0$ in $W_2$ is expected and unsurprising.

Moreover, although the authors claim that the hidden constants are dimension-free, in reality they depend exponentially on the various assumptions, so the quantitative bounds seem quite poor. Therefore, it is unclear what this adds to the literature.

In summary, the novelty of this submission does not meet the bar for acceptance, as all of the ingredients of this paper are already well-established in the literature, and the poor scaling of the bounds with problem parameters makes them of questionable value.

[1] H. Lee, J. Lu, Y. Tan, Convergence for score-based generative modeling with polynomial complexity.

**Reviewer Concerns:**

Some of the concerns regarding the exposition have been partially addressed, although there is still room for improvement.

The concerns regarding novelty remain, see above.

**Reviewer Scores:**

Reviewers Q1jr and fxPs gave poor scores and raised issues of exposition. The authors made an effort to address these concerns, and these reviewers may have increased their scores slightly.

Reviewer oCNt gave the paper a good score and likely would have kept the score unchanged.

Reviewer Gq2n engaged in a discussion with the authors and some of the concerns regarding specific technical points were addressed, but the reviewer's support remains weak and borderline at best.

---

### Decision · Program_Chairs · 2026-01-26

Reject